# A Multianalytical Approach for the Characterisation of Materials on Selected Artworks by Monogrammist IP

**DOI:** 10.3390/ma16010331

**Published:** 2022-12-29

**Authors:** Radka Šefců, Štěpánka Chlumská, Václava Antušková, Daniel Vavřík, Ivana Kumpová, Václav Pitthard

**Affiliations:** 1National Gallery Prague, Staroměstskénáměstí 12, 110 15 Prague, Czech Republic; 2Institute of Theoretical and Applied Mechanics, Czech Academy of Sciences, Prosecká 809/76, 190 00 Prague, Czech Republic; 3Kunsthistorisches Museum Wien, Burgring 5, 1010 Wien, Austria

**Keywords:** wooden artworks, Monogrammist IP, material analysis, computed tomography, glaze layer

## Abstract

This paper presents an investigation of wooden artworks from the collection of the National Gallery Prague created by Monogrammist IP–one of the top carvers of the Salzburg-Passau region at the beginning of the 16th century. His wood reliefs were examined to gain a better understanding of the historical techniques used in medieval art workshops. The internal structure of the small relief Visitation was analysed using computed tomography. Tomographic reconstruction made it possible to distinguish wood species, observe the internal structure of the artwork in detail, study the technological procedures and identify earlier repairs, additions and damages. Tomographic investigation proved the use of four types of wood on the relief Visitation, most likely pear, lime, unspecified softwood and other different species used for joining dowels. A combination of non-invasive and micro-destructive analytical techniques was employed for the chemical characterisation of the materials in the surface layers of the artworks. Photomicrographs of the surface were taken to provide material for the initial investigation. Non-invasive material research was conducted using a portable X-ray fluorescence analyser and, in selected cases, an external reflection infrared spectrometer. The detailed analyses on the micro-samples was carried out by optical microscopy, micro-Raman spectroscopy, Fourier transform infrared spectroscopy, scanning electron microscopy coupled with energy dispersive X-ray spectrometry and gas chromatography with mass spectrometry. A glaze layer based on protein with earth pigment was identified on the relief Christ the Saviour from Death.

## 1. Introduction

This paper presents the results of the material investigation of wooden artworks from the collection of the National Gallery Prague created by one of the most remarkable carvers of the Salzburg-Passau region of the first third of the 16th century, generally referred to as Monogrammist IP. The small and fragile relief of Visitation (Passau, after 1521, Inv. No. P 5197, Figure 1) is one of the few works bearing the signature “IP” (Figure 2) [1,2,3,4,5]. The artist’s monogram appears in the middle bottom part of the panel. The relief was created as a cabinet masterpiece. Its composition is inspired by sheet B. 84 (Visitation) from Albrecht Dürer’s series of woodcuts depicting Marian themes (1501–1511). It is very detailed, testifying to the artist’s mastery of relief-carving techniques. On a very small surface, the Visitation scene unfolds in several spatial planes; the artist conveyed the impression of space through perspectival foreshortening of the architectural frame and the thoughtful distribution of high and low relief. The characteristic devices of the Danube School style include the dynamic interpretation of the drapery and details of the landscape, which appear in other carvings by Monogrammist IP and his circle. There are many parallels to the Monogrammist IP’s works in paintings, drawings and prints by other leading representatives of the Danube School, namely Wolf Huber and Albrecht Altdorfer. Similar stylistic features can be observed in not only small-format carvings but also larger artworks. Zlíchov Altarpiece—another piece by Monogrammist IP from the collection of the National Gallery Prague subjected to comparative research—belongs among these types of artworks. The Zlíchov Altarpiece is composed of a central part depicting Christ the Saviour from Death (Passau, before 1526, NGP, inv. no. P 4673), while the other parts feature the Judgement Day, Christ on the Mount of Olives, Christ Appears to the three Maries, Penitent St Jerome, St Francis Receiving the Stigmata, St John the Evangelist, St Luke the Evangelist, St Matthew the Evangelist and St Mark the Evangelist (Roman Catholic Parish in Zlíchov loaned to the National Gallery Prague, Inv. Nos. P 4673–7, VP 772–6) [3].

Works of this type are particularly fragile—their condition needs to be continuously monitored and carefully documented. The dimensions of the relief are only 148 × 122 mm. The goal of the examination of this artwork was to clarify the original mounting, which consisted of a relief, frame and backing board, and the material composition of the artwork, including the possibility of the originally applied surface monochrome glaze layer. Was the Visitation originally polychromed, or did the artist coat it with a specific surface finish, described in German professional literature as Holzsichtigkeit [6]? This surface treatment, consisting of a thin monochrome glaze layer applied directly to the surface of the wood, is rarely preserved in artworks [7,8,9,10,11,12].

The beautiful bust of Mary Magdalene from the altar retable in the Ulm Minster, created by Michel Erhart and his workshop in the 1470s, is usually mentioned as the earliest example of intentionally non-polychromed carving [13,14]. Art historical literature cites many other examples of this practice, which was popular especially at the turn of the late Gothic and Renaissance. This paper discusses material and technological aspects of this practice, but it also touches upon the interpretation of its symbolism [15]. On the bust of Mary Magdalene mentioned above, the honey-coloured tone of the monochrome glaze is due to a dye acquired from the dyer’s mulberry (*Chlorophora tinctoria*, or *Maclura tinctoria*), Old Fustic, and the smoke tree (*Cotinus coggygria*, or *Rhus cotinus*), Young Fustic [16]. The literature also describes glaze materials based on proteins, polysaccharides, and plant gums pigmented with carbon black, ochres, and white inorganic pigments, or organic dyes such as yellow lake (Saffron, Berberis wood) and red lake (Lacca and Brazil wood) [17,18,19,20]. Our knowledge of the chemical composition of glazes is based on historical recipes and materials commonly used in medieval art [21,22].

We may assume that artists used plant gums, especially gum Arabic, cherry and almond gum, and resins such as mastic, sandarac and Dragon’s Blood. Storax [9] was identified in the statue of the Seated Virgin Mary by the Master of the Kefermarkt Altar (between 1475 and 1480, inv. no. P 8879) from the collection of the National Gallery Prague. Here, the resinous component is based on cinnamic acid, cinnamyl alcohol and triterpenoid resins (oleanolic and oleanenoic acids) [19,23]. The most prominent work of this master, who was a follower of Nicolaus Gerhaert from Leyden, is the unpolychromed altarpiece preserved in the church of St Wolfgang in Kefermarkt from 1485–1490 [17]. In this carving, however, the monochrome glaze layer was irreversibly removed [23,24].

Within the oeuvre of Monogrammist IP and his workshop, the monochrome surface glaze layer is preserved on the Altar of St John the Baptist from the Church of Our Lady before Týn in Prague. The layer is most likely composed of earths and white pigments bound with glair [11]. A glaze layer is also visible in a larger work, Christ the Saviour from Death. In some parts of this carving—Christ’s chest, cloaks, and vegetation—the wooden surface is visibly coloured. A combination of earth pigments and a protein binder was identified in this surface layer [25]. The specialised analytical techniques detected no organic colourants (yellow or red lake). The colour of the glaze results from subtle particles of inorganic pigments (mostly ochres) dispersed in an organic protein-based binder. As in other carvings by Monogrammist IP, the monochrome glaze layer is complemented with local polychromy, usually applied on the eyes and lips [11]. In these areas, the artist used other inorganic-based pigments, such as vermilion, ochre, carbon black, azurite, and lead white.

The Visitation panel has been preserved in its original frame, whose intricate construction makes it impossible for us to use classical documentation techniques for assessing its composition and the materials and techniques used. Therefore, micro CT was employed to determine the internal structure of the entire Visitation relief. The investigation focused on clarifying the method of the original surface treatment. The analyses were performed using non-invasive and non-destructive analytical methods. Photomicrographic documentation, X-ray fluorescence analysis, and mobile infrared spectroscopy were employed to examine the Visitation and all parts of the Zlíchov Altarpiece. Micro-samples of surface polychromy (primarily glaze layer) could only be taken from the relief of Christ the Saviour from Death (the central part of the Zlichov Altarpiece, Figure 3). The detailed analyses of the micro-samples were carried out by micro-destructive analytical techniques–optical microscopy, micro-Raman spectroscopy, Fourier transform infrared spectroscopy, scanning electron microscopy coupled with an energy dispersive X-ray spectrometry and gas chromatography with mass spectrometry. The combination of all techniques was crucial for the accurate characterisation of the structure and the identification of pigments and binding media used in the surface glaze layer.

## 2. Methods

All artworks were analysed using optical microscopy and X-ray fluorescence analysis. The small relief Visitaiton was explored via computed tomography. For other artworks, this method was not possible due to their dimensions and fragility. Reflectance infrared spectroscopy was selectively applied to the Visitation relief and St Francis Receiving the Stigmata. Other artworks were not suitable for this method because of their carved surface. Samples were obtained from relief Christ the Saviour from Death.

### 2.1. Computed Tomography

Computed tomography (CT) was performed on the small relief Visitation in the X-ray tomography laboratory of the Institute of Theoretical and Applied Mechanics of the CAS, in the centre of Telč, using TORATOM (twinned orthogonal adjustable tomograph) [26]. During tomographic measurement, the object under examination is placed vertically on the TORATOM’s rotating platform and the geometric magnification is adjusted before the measurement by setting the distances between the tube–object–detector. For this TORATOM work, X-ray tube XWT 240 SE (X-Ray WorX) operated at 90 kpV, with a target current of 130 μA, together with a GOS XRD1622 detector (Perkin Elmer, 2048 × 2048 pixels and pixel size 0.2 mm) were used. The geometric magnification was set to 1.54, giving an effective pixel size of 130 μm. During the CT measurements, 800 projections (X-ray images) were recorded during one rotation of the stage, with each projection being an average of four one-second images. CT reconstruction was performed using VolumePlayer 6.5.4 (Fraunhofer EZRT, Fürth, Germany) and visualization was done utilizing VGstudio MAX software (Volume Graphics, v. 3.5, Heidelberg, Germany).

### 2.2. Photomicrographic Documentation

Photomicrographic documentation was performed using a digital USB microscope Dino-Lite Pro AM4113ZTFV2W, polarised visible and ultraviolet light (λ = 375 nm), 1.3 megapixel and magnification 50× and 200×. The photographs were processed in the program DinoCapture 2.0 and Adobe Photoshop.

### 2.3. Mobile X-ray Fluorescence Analysis (XRF)

The X-ray fluorescence analysis was performed using a portable NITON XL3t GOLDD instrument (Thermo Scientific, Boston, MA, USA) with a mini-X-ray source using a silver anode with a maximum voltage of 50 kV. Elements heavier than aluminium (Z > 13) were detected. An integrated large-volume silicon detector was used to determine the radiation emitted from the surface of the painting from an area with a minimum diameter of 3 mm. The measured area was photographed with an integrated CCD camera. Each analysis took ca. 330 s. The measurements were taken without any direct contact with the artwork from a distance of no more than approximately 5 mm.

### 2.4. Mobile Reflectance Infrared Spectroscopy (RE-FTIR)

Non-invasive structural analysis by infrared spectroscopy was performed from the surface using a Bruker ALPHA spectrometer (Bruker, Ettlingen, Germany). Measurements were performed in the range of 8000–400 cm^−1^ with a resolution of 4 cm^−1^, with 256 scans performed. The background was measured before each sample. The spectra were evaluated in Opus.

### 2.5. Sampling

Micro-samples of polychromy could only be taken from Christ the Saviour from Death. The first micro-samples were fixed in methyl methacrylate resin (Clarocit, Struers GmbH, Willich, Germany). When hardened, the cross-section was ground and polished and subsequently analysed in a non-destructive manner. The last sample of the surface glaze was taken for the GC-MS analysis. In the case of this sample, the investigation focused solely on the surface glaze layer.

### 2.6. Optical Microscopy

Documentation of the cross-sections was carried out on a Zeiss Axio Imager Z2m light microscope (Oberkochen, Germany) in reflected light, in a dark field and after excitation by UV light (λ = 385 nm, λ = 475 nm). Usual magnification was 200–400×. The cross-sections were documented using digital camera Axiocam 712 color and the images were processed in the ZEN core v3.1.1 and Adobe Photoshop.

### 2.7. µ-Raman Spectroscopy

The cross-sections were analysed by µ-Raman spectroscopy. The Raman spectra were collected by the confocal Raman microscope Nicolet DXR (Thermo Scientific, Waltham, MA, USA) in combination with an Olympus microscope with lenses 10×, 20×, 50× and 100×, equipped with a CCD camera for signal detection. The measurement took place in the range of 3300–3350 cm^−1^. The spectral resolution was 4 cm^−1^. The diode laser with a wavelength of 780 nm was used as the excitation source. The output of the laser depended on the composition of the mixtures of pigments and the sensitivity of the individual components of the layers. The time of measurement was 1–3 min with the laser output from 0.5 to 10 mW. The measurement was performed directly on the pigment particles. The spectra were evaluated in the Omnic 9 program and compared with the spectral library.

### 2.8. Fourier Transform Infrared Spectroscopy (FTIR)

Analysis by attenuated total reflection (ATR) infrared spectroscopy was performed using an ALPHA Bruker spectrometer with a single bounce diamond crystal. The measurements were performed in the spectral range of 4000–400 cm^−^^1^ with a resolution of 4 cm^−^^1^ in 64 scans. The background (64 scans) was measured before each sample. The spectra were processed in OPUS (Bruker, Ettlingen, Germany).

### 2.9. Scanning Electron Microscope with Energy Dispersive Spectroscopy (SEM-EDS)

The X-ray microanalysis (SEM-EDS) was carried out using scanning electron microscope with an energy dispersive X-ray analyser JEOL 6460 (JEOL Ltd., Tokyo, Japan) equipped with a Si (Li) X-ray detector. The analysis was conducted under a pressure of 35 Pa and acceleration voltage of 20 kV using a backscattered electron detector. The measurements were carried out in the so-called low vacuum technique, which does not require application of a graphite coating on the sample.

### 2.10. Gas Chromatography–Mass Spectrometry (GC-MS)

The sample of the glaze layer was analysed for the presence of both proteinaceous and polysaccharide binding media by GC-MS. The analytical procedure for the analysis of proteinaceous materials is based on an acidic hydrolysis of proteins to liberate amino acids, followed by the derivatisation and quantitative determination of amino acids as their silyl derivatives. The sample was placed in conical Reacti-vial and treated with 6M hydrochloric acid (HCl, 100 µL). The sealed vial was heated to 105 °C for 24 h, removed from the heat and cooled to room temperature. The content was evaporated to dryness. High-purity water (40 µL) was added, stirred and the content was again evaporated to dryness. Ethanol (40 µL) was added twice, stirred and the content evaporated to dryness. The dried sample was then processed with a pyridine–pyridine hydrochloride mixture (10 µL) and a silylation reagent (MTBSTFA, 10 µL) and kept at 60 °C for 1h. After cooling 1 µL of the reaction mixture was injected into a GC inlet at a temperature of 300 °C. GC-MS analyses were performed using a 6890 N gas chromatograph connected to a quadrupole mass spectrometer (model 5973N), Agilent Technologies, USA. Separation was accomplished on a DB-5 MS poly (5% phenyl-95% methylsiloxane, J&W Chemical Co Ltd, Levittown, PA, USA) capillary column. The temperature of the oven was programmed from 80 °C (1 min) to 260 °C (12 min) at 6 °C min^−1^. Helium (purity 99.999%) was used as a carrier gas at an inlet pressure 100 kPa and flow 1.5 ml min^−1^.

The analytical procedure for the analysis of polysaccharides is based on an acidic hydrolysis of polysaccharide material to liberate monosaccharides followed by their transformation into oximes (to eliminate monosaccharides multiple peak production) and their subsequent derivatisation by acetic anhydride into methyloxime acetates. The sample was first treated with 1.2M trifluoroacetic acid (TFA, 100 μL) and heated to 125 °C for 1 h and then removed from the heat and cooled to room temperature. The content was evaporated to dryness in a process similar to that for proteins. Derivatisation procedure was then processed with a methoxyamine hydrochloride in pyridine mixture (60 μL) and kept at 70 °C for 10 min to form monosaccharide oximes. After cooling acetic anhydride (30 μL) was added into the mixture and heated to 70 °C for another 10 min. The final methyloxime acetates were extracted by chloroform (100 μL) and 1 μL of the extracted reaction mixture was injected into a GC inlet at a temperature of 240 °C. DB-WAX J&W Chemical Co Ltd, Levittown, PA, USA) capillary column was used for the separation of polysaccharides. The temperature of the oven was programmed from 150 °C (1 min) to 235 °C (5 min) at 10 °C min^−1^. Helium (purity 99.999%) was used as a carrier gas at an inlet pressure 100 kPa and flow 1.5 ml.min^−1^.

## 3. Results and Discussion

### 3.1. The Visitation Relief Construction, Materials and Technology

The high-resolution documentation provided precise data on the original mounting and construction of the relief panel as well as data regarding the surface and depth of the carving (Figure 4a), including evidence of original and secondary interventions and colour modifications. As evident from the visual assessment, the Visitation relief is set in a moulded frame with a backing board.

The CT images show clearly that the panel is composed of six planks with widths of 23.2–19.9–20.4–24.4–20.0–14.1 mm. The thickness of the relief panel is 1.7 mm at its thinnest point and 9.2 mm at its thickest point. Three planks are recognisable in the relief The Judgement of Paris (Monogrammist IP, ca. 1530, pear wood, 23.5 × 17.9 cm, depth 2.7 cm, Victoria and Albert Museum, London, acc. no. 4528–1858) [27]. The individual planks in the Visitation are attached by glued butt joints so precisely that they are difficult to identify with the naked eye. The micro-CT technique has made it possible to tell them apart. They are placed in a traditional way, which helps minimise the subsequent warping of the wood and ensures smoothness across the entire surface by alternating the direction of the annual rings of the adjacent planks. This layout is common in larger panels. In the Visitation, which is small and unlikely to warp, this is a sign of high precision. In the studied period, animal glues or adhesives based on casein (milk protein) were commonly used. Thanks to detailed micro-CT imaging, we can observe an irregularity in the wood—a cavity on one of the planks in both horizontal and vertical sections (Figure 5).

The relief panel is attached to the left and right frame rails by four small, slightly conical wooden dowels with a diameter of approximately 1.5 mm and 9 mm in length. The case has three moulded frame rails on the left, right and upper edge, while at the bottom, the profile of the raised edge of the relief is simpler. An analogous, smooth and unadorned frame profile appears in the relief The Judgement of Paris from Victoria and Albert Museum, London, and The Fall of Man (Monogrammist IP, ca. 1520–1530, pear wood, 16.5 cm × 13.5 cm, depth 1.4 cm, Kunsthistorisches Museum, Vienna, inv. no. KK 3984). On the back side, the Visitation is covered with a smooth backing board with a thickness of approx. 3.5 mm. There is a vertical crack on the right side of the backing board. The micro-CT sections show different densities and patterns in the wood, suggesting that a few different wood species were used. The reverse side of the case (backing board) is made of wood with a pattern that differs from the pattern in the relief panel. According to art historical literature, pear wood was a traditional material for relief carving. This wood is mentioned in connection with other extant cabinet works signed by this master. In addition to the artworks discussed above, these include the relief Lamentation (Monogrammist IP, ca. 1525, pear wood, 19.5 cm × 15.5 cm, the State Hermitage Museum in St. Petersburg, Inv. No. H.ck-1540). However, the wood from the Prague relief has never been examined by analytical methods because sampling would disturb the work’s integrity. Therefore, the micro-CT images were compared to reference images of pear (*Pyrus* sp.) and lime wood (*Tilia* sp.), which artists commonly used for small relief works, along with wild service tree (*Sorbus torminalis*) and boxwood (*Buxus* sp.). The comparison supports the assumption that Visitation was made of pear wood. This wood shows a distinctly uniform pattern of alternating annual rings. Another type of wood, probably lime, which has a lower density, was used for the relief’s backing board and the bottom rail of the frame. The micro-CT images show similar density and structure of the wood in these parts. The third type of wood was observed in the moulded frame rails on both sides and the top edge. In the annual rings, there is a marked difference in the density of the alternating late- and earlywood, characteristic of conifers. The connecting dowels were made of a yet different kind of wood.

The images also show the method of affixing the backing board to the reverse of the relief—the adhesive material was applied at least two times. Given the shape of the glue mass, we assume that the gluing took place at different times. On the CT scan, the glue appears as light-coloured circular formations. In one of the images, we can also observe how the glue was applied: there is a thin line between the two points. In one place, the glue is applied only to the corners of the panel; in another place, it appears in the corners and the middle near the top edge (Figure 4b). The gluing process indicates that the mount was previously disassembled and reassembled. This may have caused the defect in one of the connecting dowels between the relief and the frame.

### 3.2. Surface Polychromy Layer

The surfaces of the reliefs were examined using a microscope under visible and UV light, XRF and RE-FTIR. Currently, the relief Visitation has no polychromy. White relics of polychromy are visible under UV light due to its fluorescence in places, usually in difficult-to-access depths of relief, in which the presence of chalk remains was confirmed by elemental XRF analysis. The investigation showed that the Visitation relief was secondarily polychromed in the past. The polychrome layer was later carefully removed, as evidenced by the almost intact condition of the fine details in the carving. This practice is not unique within the surviving oeuvre of the Monogrammist IP. Similar fragments of white polychromy were also observed in some parts of the Zlíchov Altarpiece. The XRF analysis detected calcium, zinc, lead, iron and sporadically copper in the retable Judgement Day and wings with St Mark the Evangelist, St Luke the Evangelist and St John the Evangelist (Figure 6).

These elements may originate from white pigments, namely chalk (calcium carbonate), lead white or zinc white, in secondary polychrome layers. Iron, characteristic of earth pigments, was also detected in some places. On the Visitation relief, the amount of iron is negligible. In the past, earth pigments were mostly used to adjust the colour of the wood surface. Earth pigments were identified on other artworks attributed to Monogrammist IP. Earths with white pigments and glair-based binder were found in the Altar of St John the Baptist from the Church of Our Lady before Týn in Prague [11].

To observe the character of the monochrome glaze layer, we performed an analysis of the sample from the central part of the Zlíchov Altarpiece, Christ the Saviour from Death, made of lime wood. The wood species was identified based on the presence of spiral thickenings in the vessels which are characteristic of lime wood (Figure 7). On the cross-section, an organic layer is visible under the fragment of secondary white polychromy (Figure 8). In the white layer, calcium carbonate (chalk) was identified by Raman spectroscopy based on characteristic band at 1085 cm^−1^. This pigment was confirmed by FTIR as well. Besides calcium from calcium carbonate, EDS analysis proved the presence of zinc characteristic of zinc white. Locally, the glaze layer contained magnesium, aluminium, silicon and iron, suggesting the presence of earth pigments.

Iron, aluminium and silicon were identified in the fragments of the darker glaze layer, corresponding to the use of earth pigments common in the period practice. The GC-MS analysis of the sample without the secondary white layer confirmed the use of a protein-based binder in the glaze layer (Figure 9). The proteinaceous binder was impossible to specify more precisely due to an incomplete profile of amino acids. Non-invasive RE-FTIR analysis of the surface layers on the Visitation and St Francis Receiving the Stigmata relief confirmed the presence of wax (Figure 10) that was identified based on absorption bands at ~ 4320, 4251, 2925, 2855, 1740 cm^−1^ and doublets at 1475, 1465 and 730, 720 cm^−1^. The wax was applied secondarily to protect the surface. No other organic compounds were identified using non-invasive analytical techniques.

## 4. Conclusions

The goal of the examination was to clarify the mounting and the original layer applied on the surface of the relief Visitation and to explore the surface glaze layers on other artworks by Monogrammist IP from the collection of the NGP. The composition of the individual parts of the Visitation relief and its construction were documented using high-resolution computed tomography. The documentation of the individual parts of the frame showed that it was repaired a few times in the past. During these repairs, the frame was probably disassembled. A different type of wood was used for one of the frame rails, suggesting that this rail is a later addition. The bottom rail was made of a different wood (probably lime wood) than the relief, which was very likely carved in pear wood. The other moulded rails are made of coniferous wood. The comparison of the Visitation relief with other works by Monogrammist IP, such as The Judgement of Paris from the Victoria and Albert Museum and The Fall of Man from the Kunsthistorisches Museum, Vienna, shows that there are no similarities between these works and the Prague carving. We cannot rule out that the existing frame comes from a later period. However, it is also possible, based on the profile of the frame rails that the frame was made at the same time as the relief. Older records suggest that in this form, the artwork was part of the Waldstein family collection, from where it came to the state collection in 1948. Monochrome glaze layers are difficult to analyse using non-invasive imaging and analytical techniques. Therefore, a monochrome glaze layer has not been unambiguously confirmed in the Visitation relief. There are only remnants of chalk from secondary polychromy, which was later removed. Analysis of polychrome samples from Christ the Saviour from Death confirmed the use of calcium carbonate and zinc white in the secondary white layer. The glaze layer in this work consists of earth pigments mixed with proteinaceous binder. On other artworks, a protective wax layer was identified on the surface.

## Figures and Tables

**Figure 1 materials-16-00331-f001:**
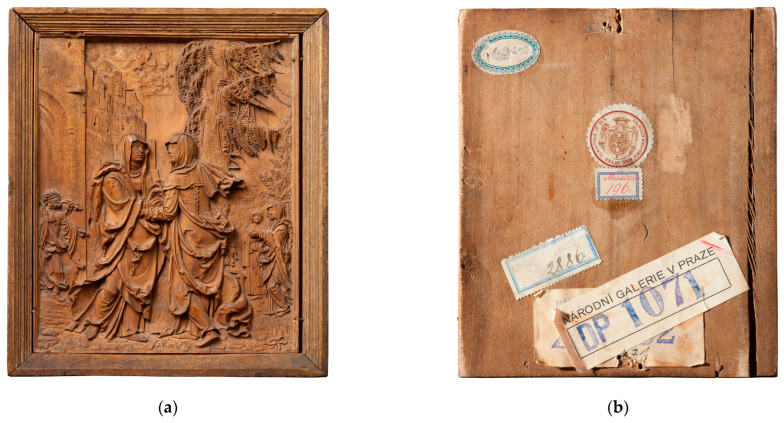
Monogrammist IP, Visitation, averse (**a**) and reverse (**b**).

**Figure 2 materials-16-00331-f002:**
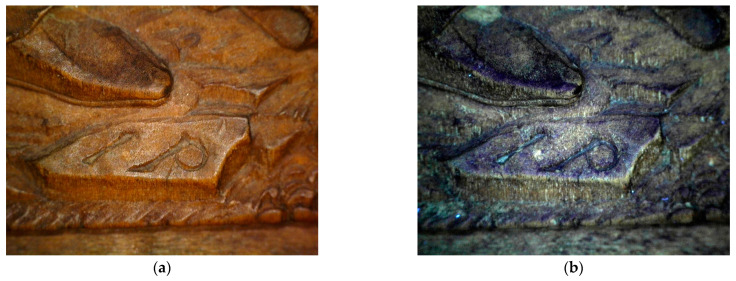
Photomicrographs in VIS (**a**) and UV light (**b**) of the monogramme.

**Figure 3 materials-16-00331-f003:**
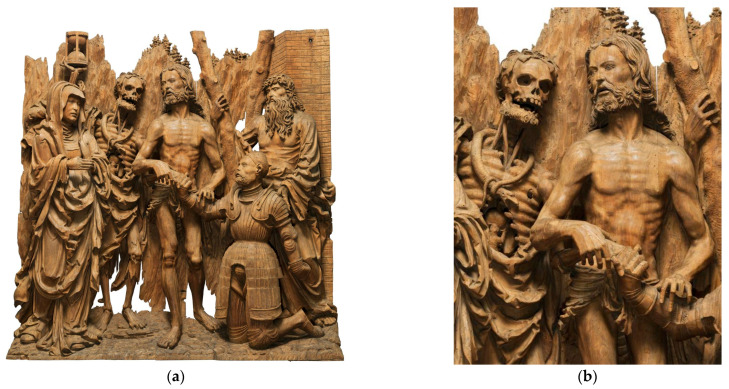
Monogrammist IP, Zlíchov Altarpiece, central part featuring Christ the Saviour from Death (**a**) and a detail with monochrome glaze on the Christ’s chest and black eyes (**b**).

**Figure 4 materials-16-00331-f004:**
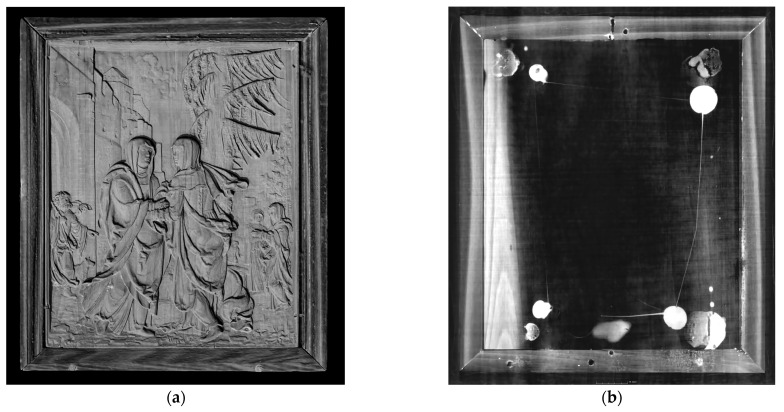
Micro-CT images of the surface of the Visitation relief (**a**) and plane section of the relief at the junction of the backing board and the back of the relief with visible traces of glue (**b**).

**Figure 5 materials-16-00331-f005:**
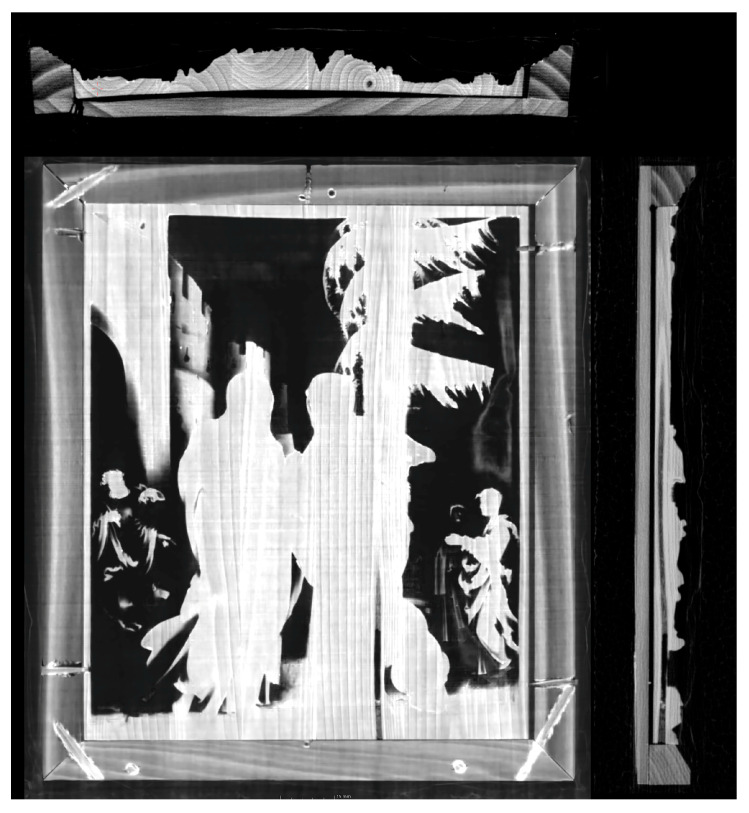
Micro-CT images of the Visitation relief: horizontal and vertical sections, section in the plane of the relief with a visible defect in the wood, fastening elements and difference in the wood patterns.

**Figure 6 materials-16-00331-f006:**
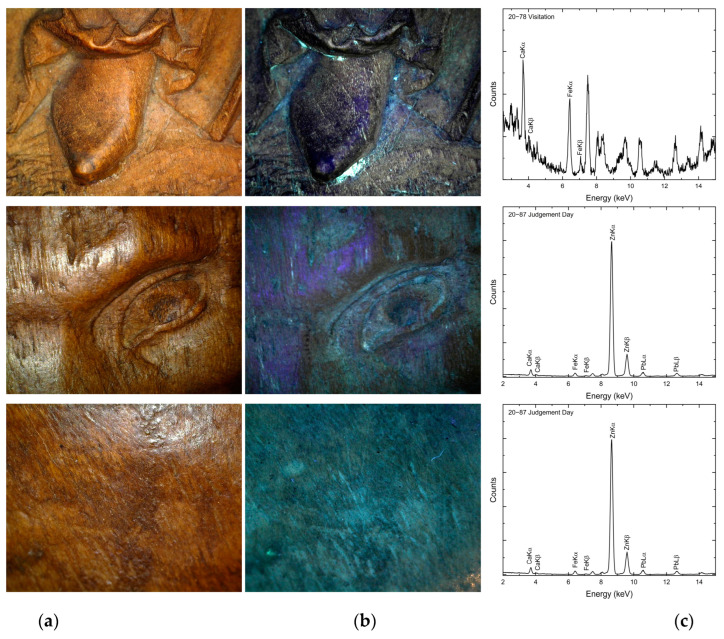
Photomicrographic documentation of the surface in visible (**a**) and UV light (**b**) with a fragment of secondary polychromy and corresponding XRF spectra (**c**); from the top: Visitation, St Luke the Evangelist and Judgement Day.

**Figure 7 materials-16-00331-f007:**
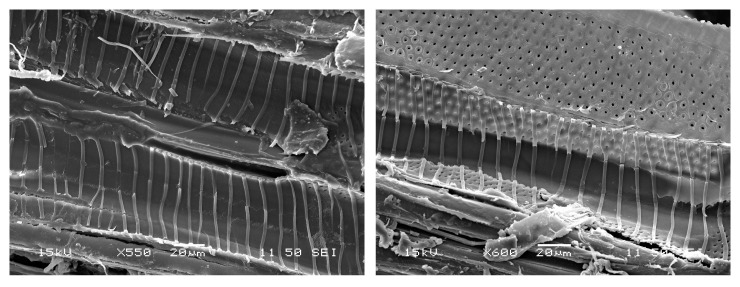
Secondary electron images of the sample of the lime wood from the carving Christ the Saviour from Death.

**Figure 8 materials-16-00331-f008:**
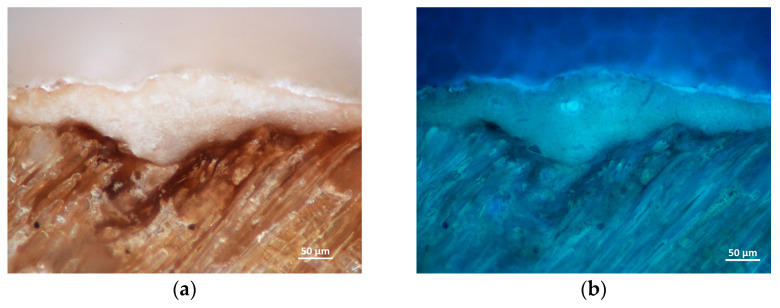
Cross-section of the sample from the surface with white polychromy from the carving Christ the Saviour from Death in visible (**a**) and UV light (**b**) and EDS spectra of the white layer (**c**) and glaze layer (**d**).

**Figure 9 materials-16-00331-f009:**
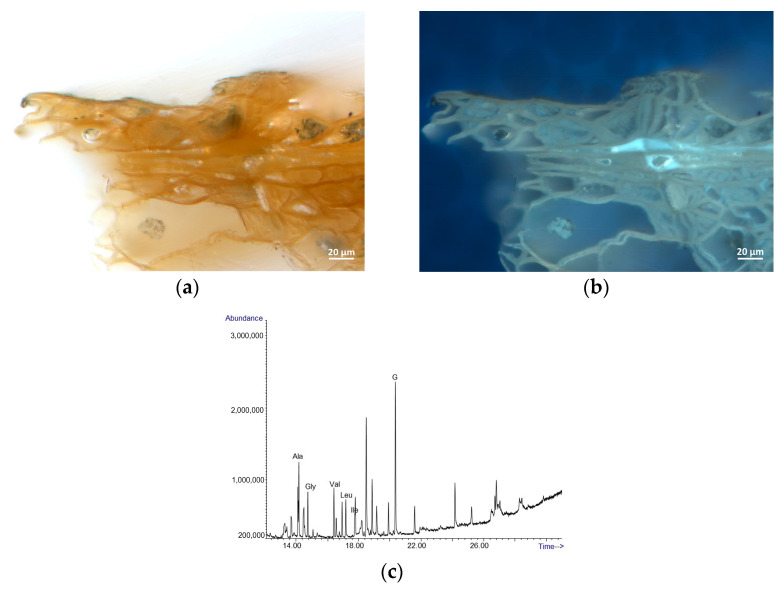
Cross-section of the sample from the surface with glaze layer from the carving Christ the Saviour from Death in visible (**a**) and UV light (**b**) and TIC chromatogram of the organic materials from the glaze layer (**c**). Note: G = glycerol and amino acids from protein (Ala = alanine, Gly = glycine, Val = valine, Leu = leucine, Ile = isoleucine).

**Figure 10 materials-16-00331-f010:**
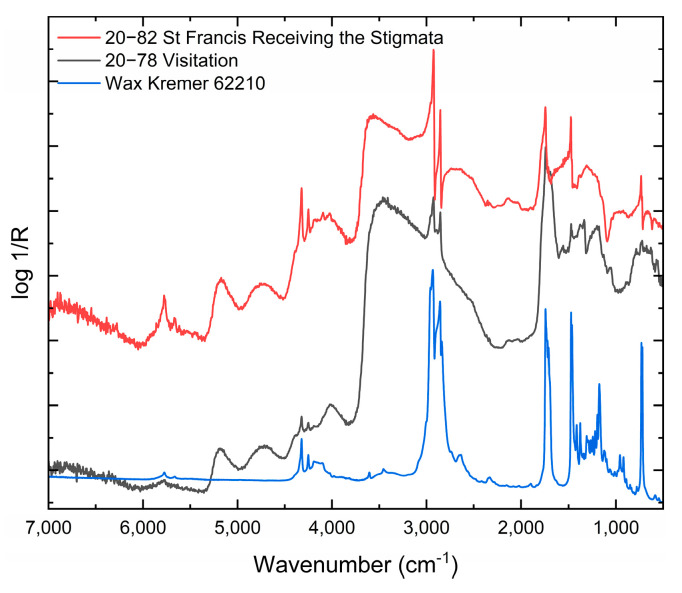
RE-FTIR spectra from the surfaces of reliefs St Francis Receiving the Stigmata and Visitation and corresponding spectrum of wax.

## Data Availability

Not applicable.

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
