# Peer review of "A Multianalytical Approach for the Characterisation of Materials on Selected Artworks by Monogrammist IP"

_materials, 2022, doi:10.3390/ma16010331_

Round 1
Reviewer 1 Report
- Whole the manuscript should be completely checked with native expert.
- The main achievements of the project should be brought at the end of the Abstract section.
- All of the references should be write in same style:
- For example ref No. 17, was written with no pp but, ref No. 18 was written in other style.
- The energy dispersive technique and its results did not explain and discuss well.
- How did UV light examination assist the authors for their investigations?
- Which JCPDS No. cards did authors used for XRD evaluation?
- FTIR analysis did not discuss well in Fig. 10.
Author Response
We would like to thank you for your valuable comments and for the advice that were used for manuscript correction.
Reviewer 1
Whole the manuscript should be completely checked with native expert.
Responses on comments: The manuscript was checked entirely by a professional.
The main achievements of the project should be brought at the end of the Abstract section.
Responses on comments: Added to text. Changes are highlighted in blue.
All of the references should be write in same style. For example ref No. 17, was written with no pp but, ref No. 18 was written in other style.
Responses on comments: References were corrected and formatted according to the journal's requirements
The energy dispersive technique and its results did not explain and discuss well.
Responses on comments: Added to text. Changes are highlighted in blue.
Jak pomohlo vyšetření UV světlem autorům při jejich zkoumání?
Reakce na komentáře: Zbytky sekundární polychromie byly jasně viditelné pod UV světlem díky jeho fluorescenci. Do textu bylo přidáno vysvětlení. Změny jsou zvýrazněny modře.
Které karty JCPDS č. autoři použili pro hodnocení XRD?
Odpovědi na komentáře: XRD analýza nebyla provedena.
FTIR analýza na obr. 10 dobře nediskutovala.
Reakce na připomínky: Identifikace byla založena na přítomnosti charakteristických absorpčních pásů, které byly přidány do textu. Změny jsou zvýrazněny modře.
Reviewer 2 Report
The authors did great job analyzing the materials from selected artworks by Monogrammist IP. The approaches they employed are optical microscopy, micro-Raman spectroscopy, Fourier transform infrared spectroscopy, scanning electron microscopy coupled with energy dispersive X-ray spectrometry and gas chromatography with mass spectrometry. The experimental section and conclusion section are detailed and data supported. I suggest publishing gafter minor revision.
1. Please provide the instrument information for GC-MS (brand, type, etc.).
2. Please provide the flow rate and temperature gradient when running GC-MS.
3. What are the major peaks between Ile and G in Figure 9?
Author Response
We would like to thank you for your valuable comments and for the advice that were used for manuscript correction.
Please provide the instrument information for GC-MS (brand, type, etc.).
Responses on comments: Added to text. Changes are highlighted in green.
Please provide the flow rate and temperature gradient when running GC-MS.
Responses on comments: Added to text. Changes are highlighted in green.
What are the major peaks between Ile and G in Figure 9?
Responses on comments: Due to the minute-size of the sample, the intensities of amino acids ‘peaks belonging to organic binder are relatively small in comparison to the unwanted major peaks from the background, which significantly interfered and altered the whole chromatogram of the sample. Nevertheless, the chromatogram is shown in the text despite of this effect.
Reviewer 3 Report
The issue addressed in the paper discusses the investigation of wooden artworks from the collection of the National Gallery Prague created by Monogrammist IP. As the Authors report, material investigation of wood reliefs was performed to understand the historical techniques used in the medieval art workshops.
First of all, I find that an important topic, compatible with the journal's scope, was considered.
Such studies are partially analysed in literature. It would be worth presenting the state of the art in a broader way. I suggest a more dilligent, comparative description of other scientific research from the literature (for example, it is possible to add a short state of the art comparative analysis report / section 1).
I also recommend several corrections to improve the quality of this paper:
- to precisely define the research scenario (it is very general); needed to clarify the scope of the study and consequently a clear, step-by-step, simple, synthetic research pattern; yes, the methodology is described, but I recommend more precision, as the reader should know how to repeat a similar analysis on this basis (please consistently correct and complete section 2);
- to briefly explain whether there is need to use, for instance, other methods;
- to improve the readability and description of tables and figures (since they are the basis for analysis verification), supplement the history of their description, a clear and not laconic reference in the paper (in section 3, especially for figures 6 - 10).
Please remember that the formulated objectives - find a clear answer in the conclusion of the study. Is this really the way it works? Does the conclusion answer all the questions posed at the beginning of the paper (expressed in objectives and hypotheses)? Please complete it and also correct it.
There is no section 4 in the paper , but there is a section 5. Has something been forgotten?
The results discussion is laconic. It is difficult to link it to the main objective, to the specific objectives of the study, to a possible verification of the hypotheses? The methodological area of the study needs, in my opinion, to be supplemented.I also strongly suggest that recommendations for specific, practical, not only general (and not entirely clear) applications of this research shall be provided (section 5).
The language of this paper is relatively correct, however some descriptions would benefit from being more concise. I recommend that the authors cooperate with a native speaker to improve the text of the paper.
Author Response
We would like to thank you for your valuable comments and for the advice that were used for manuscript correction.
Such studies are partially analysed in literature. It would be worth presenting the state of the art in a broader way. I suggest a more dilligent, comparative description of other scientific research from the literature (for example, it is possible to add a short state of the art comparative analysis report / section 1).
Responses on comments: Publications concerning technological aspects of the artworks of Monogrammist IP and monochrome glaze layers are sporadic. All relevant publications gathered during the preparation of the article were mentioned in the text. References to older literature are in cited publications.
- to precisely define the research scenario (it is very general); needed to clarify the scope of the study and consequently a clear, step-by-step, simple, synthetic research pattern; yes, the methodology is described, but I recommend more precision, as the reader should know how to repeat a similar analysis on this basis (please consistently correct and complete section 2);
Responses on comments: Added to text. Changes are highlighted in yellow.
- to briefly explain whether there is need to use, for instance, other methods;
Responses on comments: Methods were chosen based on gaining the maximum amount of information focused on research goals defined in the article with regard to the nature of the artworks.
- to improve the readability and description of tables and figures (since they are the basis for analysis verification), supplement the history of their description, a clear and not laconic reference in the paper (in section 3, especially for figures 6 - 10).
Responses on comments: Added to text. Changes are highlighted in blue and yellow. Figures and their titles were corrected.
Please remember that the formulated objectives - find a clear answer in the conclusion of the study. Is this really the way it works? Does the conclusion answer all the questions posed at the beginning of the paper (expressed in objectives and hypotheses)? Please complete it and also correct it.
Responses on comments: We rearranged and added details in the section Conclusion and hope that it’s clearer now. Changes are highlighted in blue.
There is no section 4 in the paper, but there is a section 5. Has something been forgotten?
Responses on comments: Chapter numbering was fixed.
The results discussion is laconic. It is difficult to link it to the main objective, to the specific objectives of the study, to a possible verification of the hypotheses? The methodological area of the study needs, in my opinion, to be supplemented.I also strongly suggest that recommendations for specific, practical, not only general (and not entirely clear) applications of this research shall be provided (section 5).
Responses on comments: We rearranged and added the text and hope that it’s clearer now. Changes are highlighted in yellow, blue and green.
The language of this paper is relatively correct, however some descriptions would benefit from being more concise. I recommend that the authors cooperate with a native speaker to improve the text of the paper.
Responses on comments: The manuscript was checked entirely by a professional.
Round 2
Reviewer 1 Report
The manuscript is acceptable in the present form.